# SUBJECT-DIFFUSION: OPEN DOMAIN PERSONALIZED TEXT-TO-IMAGE GENERATION WITHOUT TEST-TIME FINE-TUNING

## ABSTRACT

Recent progress in personalized image generation using diffusion models has been significant. However, development in the area of open-domain and test-time fine-tuning-free personalized image generation is proceeding rather slowly. In this paper, we propose Subject-Diffusion, a novel open-domain personalized image generation model that, in addition to not requiring test-time fine-tuning, also only requires a single reference image to support personalized generation of single- or multi-subjects in any domain. Firstly, we construct an automatic data labeling tool and use the LAION-Aesthetics dataset to construct a large-scale dataset consisting of 76M images and their corresponding subject detection bounding boxes, segmentation masks, and text descriptions. Secondly, we design a new unified framework that combines text and image semantics by incorporating coarse location and fine-grained reference image control to maximize subject fidelity and generalization. Furthermore, we also adopt an attention control mechanism to support multi-subject generation. Extensive qualitative and quantitative results demonstrate that our method have certain advantages than other frameworks in single, multiple, and human-customized image generation.

## 1 INTRODUCTION

Recently, with the rapid development of diffusion-based generative models (Ho et al., 2020; Song et al., 2020b;a), many large synthesis models (Rombach et al., 2022; Ramesh et al., 2022; Nichol et al., 2022; Saharia et al., 2022; Balaji et al., 2022; Feng et al., 2023) trained on large-scale datasets containing billions of image-text pairs, *e.g.,* LAION-5B (Schuhmann et al., 2022), have shown amazing text-to-image generation ability with fantastic artistry, authenticity, and semantic alignment. However, merely textual information is insufficient to fully translate users' intentions. Therefore, integrating textual description and reference images to generate new customized images is an emerging research direction.

Based on a pre-trained text-to-image generation model, *e.g.,* Stable Diffusion (Rombach et al., 2022) and Imagen (Saharia et al., 2022), a group of approaches (Gal et al., 2022; Ruiz et al., 2023a; Kumari et al., 2023; Tewel et al., 2023; Avrahami et al., 2023; Hao et al., 2023; Smith et al., 2023) propose to fine-tune the models on each group of the provided reference images (typically 3-5 images). Although these methods yield satisfactory results, they require specialized training of the network (word embedding space (Gal et al., 2022), specific layers of the UNet (Ruiz et al., 2023a; Kumari et al., 2023) or some adding side-branches (Smith et al., 2023)), which is inefficient for realistic application. Another technique roadmap (Xiao et al., 2023; Wei et al., 2023; Chen et al., 2023c; 2022) is to re-train the text-to-image base model by specially designed network structures or training strategies on a large-scale personalized image dataset, but often results in inferior fidelity and generalization as compared with test-time fine-tuning approaches. Further, some methods can only achieve personalized image generation on specific domains, such as portrait (Xiao et al., 2023; Shi et al., 2023; Jia et al., 2023), cats (Shi et al., 2023) or dogs (Jia et al., 2023). Even though some recent proposed algorithms (Wei et al., 2023; Li et al., 2023a; Ma et al., 2023b) can achieve open-domain customized image generation, they only handle single-concept issues. With regard to a single reference image, multiple concept generation, the absence of test-time fine-tuning, and open-domain zero-shot capability, we summarize a comprehensive list of personalized image generation papers as

in Appendix C. According to the statistics, no algorithm is currently available that can fully satisfy the four conditions listed above. As a result, we are motivated to propose Subject-Diffusion, an open-domain personalized text-to-image generation framework that only needs one reference image and doesn't require test-time fine-tuning.

A large-scale training dataset with object-level segmentation masks and image-level detailed language descriptions is crucial for zero-shot customized image generation. While for such a laborious labeling task, publicly available datasets, including LVIS (Gupta et al., 2019), ADE20K (Zhou et al., 2019), COCO-stuff (Caesar et al., 2018), Visual Genome (Krishna et al., 2017) and Open Images (Kuznetsova et al., 2020), typically have insufficient image volumes ranging from 10k to 1M or even no text description. To address the data shortage for open-domain personalized image generation, we are motivated to build an automatic data labeling tool, which will be detailedly introduced in Sec. 3.1.

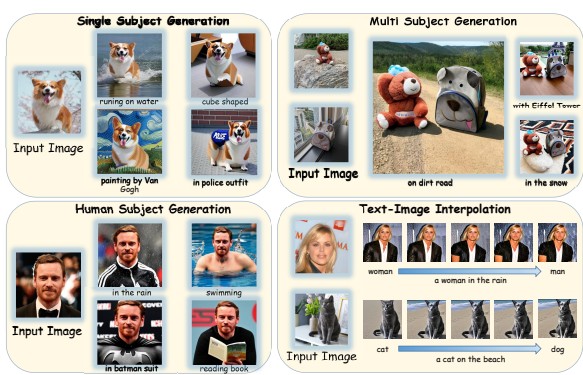

Figure 1: Our Subject-Diffusion is capable of generating high-fidelity subject-driven images (general and human subjects) using just one reference image, without test-time fine-tuning, allowing for the preservation of identity and editability. Furthermore, our model supports the generation of multiple subjects within a single model. We also show the interpolation ability between reference images and word concepts.

As mentioned in (Zhou et al., 2023), the information of personalized images may overwhelmingly dominate that of user input text to prevent creative generation. In order to balance fidelity and editability, we propose to fuse the input text prompt and object-level image features by continually training the CLIP text encoder (unlike fixing the encoder as FastComposer (Xiao et al., 2023) and Elite (Wei et al., 2023)) based on a specific prompt style. We further propose to integrate fine-grained reference image patches, detected object bounding boxes, and location masks to control the fidelity of generated images. Finally, to further control the generation of multiple subjects, we introduce cross-attention map control during training. As exhibited in Fig. 1, based on the constructed large-scale structured data in an open domain and our proposed new model architecture, Subject-

Diffusion achieves remarkable fidelity and editability, which can perform single, multiple, and human subject personalized generation by modifying their shape, pose, background, and even style with only one reference image for each subject. In addition, Subject-Diffusion can also perform smooth interpolation between customized images and text descriptions by using a specially designed denoising process. In terms of quantitative comparisons, our model have certain advantages than other methods, including test-time fine-tuning and non-fine-tuning approaches on the DreamBench (Ruiz et al., 2023a) and our proposed larger open-domain test dataset.

In summary, our contributions are threefolds: **(i)** We design an automatic dataset construction pipeline and create a sizable and structured training dataset that comprises 76M open-domain images and 222M entities. **(ii)** To the best of our knowledge, we propose a personalized image generation framework which is the first work to address the challenge of simultaneously generating open-domain single- and multi-concept personalized images without test-time fine-tuning, solely relying on a single reference image for each subject. **(iii)** Both quantitative and qualitative experimental results demonstrate the excellent performance of our framework as compared with other methods.

## 2 RELATED WORK

### 2.1 TEXT-TO-IMAGE GENERATION

The diffusion model has emerged as a promising direction to generate images with high fidelity and diversity based on provided textual input. GLIDE (Nichol et al., 2022) utilizes an unclassified guide to introduce text conditions into the diffusion process. DALL-E2 (Ramesh et al., 2022) uses a diffusion prior module and cascading diffusion decoders to generate high-resolution images based on the CLIP text encoder. Imagen (Saharia et al., 2022) emphasizes language understanding and suggests using a large T5 language model to better represent semantics. Latent diffusion model (Rombach et al., 2022)

uses an autoencoder to project images into latent space and applies the diffusion process to generate latent-level feature maps. Stable diffusion (SD) (Rombach et al., 2022), ERNIE-ViLG2.0 (Feng et al., 2023) and ediffi (Balaji et al., 2022) propose to employ a cross-attention mechanism to inject textual conditions into the diffusion generation process. Our framework is built on the basis of SD due to its flexible scalability and open-source nature.

## 2.2 SUBJECT-DRIVEN TEXT-TO-IMAGE GENERATION

Currently, there are two main frameworks for personalized text-to-image generation from the perspective of whether to introduce test-time fine-tuning or not. In terms of test-time fine-tuning strategies, a group of solutions require several personalized images containing a specific subject and then directly fine-tune the token embedding of the subject to adapt to learning visual concepts (Gal et al., 2022; Han et al., 2023a; Yang et al., 2023; Voynov et al., 2023; Alaluf et al., 2023). Another group of approaches fine-tune the generation model using these images (Ruiz et al., 2023a; Kumari et al., 2023; Han et al., 2023b; Fei et al., 2023; Chen et al., 2023a), among which DreamBooth (Ruiz et al., 2023a) fine-tunes the entire UNet network, while Custom Diffusion (Kumari et al., 2023) only fine-tunes the K and V layers of the cross-attention in the UNet network. On the other hand, Custom Diffusion proposes the personalized generation of multiple subjects for the first time. SVDiff (Han et al., 2023b) constructs training data using cutmix and adds regularization penalties to limit the confusion of multiple subject attention maps. Cones proposes concept neurons and updates only the concept neurons for a single subject in the K and V layers of cross-attention. For multiple personalized subject generation, the concept neurons of multiple trained personalized models are directly concatenated. Mix-of-Show (Gu et al., 2023) trains a separate LoRA model (Hu et al., 2021) for each subject and then performs fusion. Cones 2 (Liu et al., 2023c) generates multi-subject combination images by learning the residual of token embedding and controlling the attention map.

Since the test-time fine-tuning methods suffer from a notoriously time-consuming problem, another research route involves constructing a large amount of domain-specific data or using open-domain image data for training without additional fine-tuning. InstructPix2Pix (Brooks et al., 2023) can follow human instructions to perform various editing tasks, including object swapping, style transfer, and environment modification, by simply concatenating the latent of the reference images during the model's noise injection process. ELITE (Wei et al., 2023) proposes global and local mapping training schemes by using the OpenImages testset, which contains 125k images and 600 object classes as the training data. However, due to the limitations of the model architecture, the text alignment effect is relatively moderate. UMM-Diffusion presents a novel Unified Multi-Modal Latent Diffusion (Ma et al., 2023b) that takes joint texts and images containing specified subjects as input sequences and generates customized images with the subjects. Its limitations include not supporting multiple subjects and its training data being selected from LAION-400M (Schuhmann et al., 2021), resulting in poor performance in generating rare themes. Similarly, Taming Encoder (Jia et al., 2023), InstantBooth (Shi et al., 2023), and FastComposer (Xiao et al., 2023) are all trained on domain-specific data. BLIP-Diffusion(Li et al., 2023a) uses OpenImages data, and due to its two-stage training scheme, it achieves good fidelity effects but does not support multi-subject generation.

In contrast, our model, which is trained on a sizable self-constructed open-domain dataset, performs exceptionally well in terms of the trade-off between fidelity and generalization in both single- and multi-subject generation by providing only one reference image for each subject.

## 3 METHODOLOGY

In this section, we will first introduce our constructed large-scale open-domain dataset for personalized image generation. Then, an overview of the Subject-Diffusion framework, followed by an explanation of how we leverage auxiliary information, including reformulating prompts, integrating fine-grained image and location information, and a cross-attention map control strategy, will be detailed.

## 3.1 DATASET CONSTRUCTION

To endow the diffusion model with the capability of arbitrary subject image generation, a huge multimodal dataset with open-domain capabilities is necessary. However, the existing image datasets either have a small number of images, such as COCO-Stuff (Caesar et al., 2018) and OpenIm-

ages (Kuznetsova et al., 2020), or lack modalities (segmentation masks and detection bounding boxes) and have inconsistent data quality, such as LAION-5B (Schuhmann et al., 2022). Therefore, we are inspired to create a high-quality, large-scale multimodal dataset that is suitable for our task.

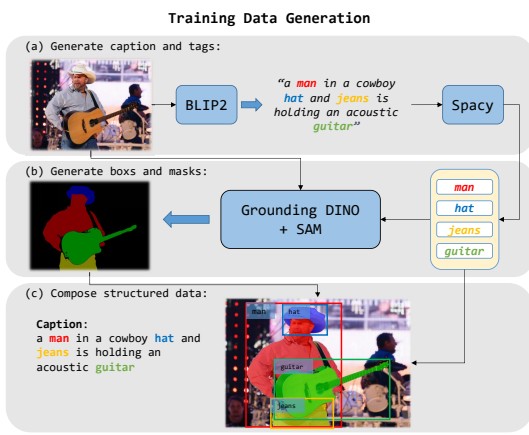

Figure 2: The procedure for training data generation.

As depicted in Fig. 2, we outline the three steps we took to create our training data based on LAION-5B. The captions for images provided by LAION-5B are of poor quality, often containing irrelevant or nonsensical wording. This can pose a significant challenge for text-to-image tasks that require accurate image captions. To address this issue, by using BLIP-2 (Li et al., 2023a), we can generate more precise captions for each image. However, for our subject-driven image generation task, we also need to obtain entities' masks and labels from the images. In order to accomplish this, we perform part-of-speech analysis on the generated captions and treat the nouns as entity tags. Once we have obtained the entity labels, we can use the open-set detection model Grounding DINO (Liu et al., 2023a) to detect the corresponding location of the entity and use the detection box as a cue for the segmentation model SAM (Kirillov et al., 2023) to determine the corresponding mask. Finally, we combine the image-text pairs, detection boxes, segmentation masks, and corresponding labels for all instances to structure the data. Based on the aforementioned pipeline, we apply sophisticated filtering strategies, as detailed in Appendix B.1, to form the final high-quality dataset called **S**ubject-**D**iffusion **D**ataset (SDD). Our dataset contains 76M examples, 222M entities, and 162K common object classes, which is much larger than the number of annotated images in the famous OpenImages (1M images) (Kuznetsova et al., 2020). Furthermore, it also covers a wide range of variations involving the capture of scenes, entity classes, and photography conditions (resolution, illumination, *etc.*). This great diversity, as well as its large scale, offers great potential for learning subject-driven image generation abilities in the open domain, which is believed to boost the development of generative artificial intelligence. Please refer to Appendix B.2 for more dataset statistics and comparisons.

## 3.2 MODEL OVERVIEW

The overall training framework of our proposed Subject-Diffusion is illustrated in Fig. 3. The design spirit of Subject-Diffusion mainly focuses on three components. First, we specifically design the prompt format and employ a text encoder to fuse the text and object-level visual features as the conditions for SD. Second, to further enhance the fidelity of the generated personalized images, we propose to insert an adapter between each self- and cross-attention block, which encodes dense patch features of the segmented objects and their corresponding bounding box information. Third, in order to endow Subject-Diffusion with multiple customized image generation abilities, we propose to employ a cross-attention map control strategy based on segmentation masks to enforce a model focusing on local optimization between the entity and its corresponding area.

## 3.3 EXPLOITATION OF AUXILIARY INFORMATION

**Fusion text encoder**. As proposed in Texture Inversion (Gal et al., 2022), a learned image embedding incorporated with prompt embedding is essential to achieving personalized image generation. Therefore, we first construct a new prompt template similar to BLIP-Diffusion (Li et al., 2023a): "*[text prompt], the [subject label 0] is [PH_0], the [subject label 1] is [PH_1], ...*" where "*text prompt*" represents the original text description, "*subject label ⁎*" represents the category label of the subject, and "*PH_⁎*" are place holders corresponding to the subject image. Then, in contrast to approaches (Ma et al., 2023a; Shi et al., 2023; Xiao et al., 2023; Ma et al., 2023b), we choose to fuse text and image information before the text encoder. We conduct extensive experiments, showing that fusing text and image information before the text encoder and then retraining the entire text encoder has stronger self-consistency than fusing them later. Specifically, we replace the entity token

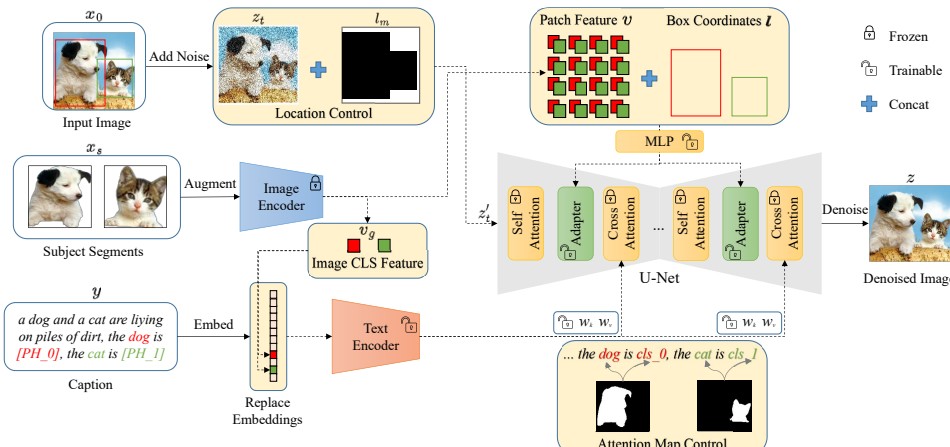

Figure 3: An overview of the proposed Subject-Diffusion method based on SD structure. **(i)** We first design a specific condition by integrating the text prompt and object image features. **(ii)** Then, we extract fine-grained image local patch features, combining with the detected object bounding boxes, and insert an adapter module between self- and cross-attention in the UNet to enhance the model's fidelity ability. **(iii)** Further, we propose to employ an attention map control strategy to deal with the multiple object generation issue.

embedding at the first embedding layer of the text encoder with the image subject "CLS" embedding at the corresponding position, and then retrain the entire text encoder.

**Dense image and object location control.** Generating personalized images in an open domain while ensuring the fidelity of the subject image with only textual input poses a significant challenge. To address this challenge, we propose to incorporate dense image features as an important input condition, similar to the textual input condition. To ensure that the model focuses solely on the subject information of the image and disregards the background information, we feed the segmented subject image into the CLIP (Radford et al., 2021) image encoder to obtain 256-length patch feature tokens. Furthermore, to prevent confusion when generating multiple subjects, we fuse the corresponding image embedding with the Fourier-transformed coordinate position information of the subject. Subsequently, we feed the fused information into the UNet framework for learning, similar to GLIGEN (Li et al., 2023b). In each Transformer block, we introduce a new learnable adapter layer between the self-attention layer and the cross-attention layer, which takes the fused information as input and is defined as $\mathcal{L}_a := \mathcal{L}_a + \beta \cdot tanh(\gamma) \cdot S([\mathcal{L}_a, h^e])$, where $\mathcal{L}_a$ is the output of the self-attention layer, $\beta$ a constant to balance the importance of the adapter layer, $\gamma$ a learnable scalar that is initialized as 0, $S$ the self-attention operator, and $h^e = MLP([v, Fourier(l)])$, where $MLP(\cdot, \cdot)$ is a multi-layer perceptron that concatenates the two inputs across the feature dimension: $v$ the visual 256 patch feature tokens of an image, and $l$ the coordinate position information of the subject. In the process of training the UNet model, we selectively activate the key and value layers of cross-attention layers and adapter layers while freezing the remaining layers. This approach is adopted to enable the model to focus more on learning the adapter layer.

In addition, to prevent model learning from collapsing, a location-area control is innovatively introduced to decouple the distribution between foreground and background regions. Specifically, as shown in Fig. 3, a binary mask feature map is generated and concatenated to the original image latent feature for a single subject. For multiple subjects, we overlay the binary images of each subject and then concatenate them onto the latent feature. During inference, the binary image can be specified by the user, detected automatically based on the user's personalized image, or just randomly generated.

**Cross attention map control.** Currently, text-to-image generation models often encounter confusion and omissions when generating multiple entities. Most solutions involve controlling the cross-attention map during model inference (Wu et al., 2023; Wang et al., 2023; Rassin et al., 2023; Chefer et al., 2023). The proposed approaches in this study are primarily based on the conclusions drawn from Prompt-to-Prompt (Hertz et al., 2022): The cross-attention in the text-to-image diffusion models can reflect the positions of each generated object specified by the corresponding text token, which is calculated from:

$$CA_l(z_t, y_k) = Softmax(Q_l(z_t) \cdot L_l(y_k)^T), \qquad (1)$$

where $CA_l(z_t, y_k)$ is the cross-attention map at layer $l$ of the denoising network between the intermediate feature of the noisy latent $z_t$ and the $k_{th}$ text token $y_k$, and $Q_l$ and $L_l$ are the query and key projections. For each text token, we could get an attention map of size $h_l \times w_l$, where $h_l$ and $w_l$ are the spatial dimensions of the feature $z_t$ and the cross-attention mechanism within diffusion models governs the layout of generated images. The scores in cross-attention maps represent the amount of information that flows from a text token to a latent pixel. Similarly, we assume that subject confusion arises from an unrestricted cross-attention mechanism, as a single latent pixel can attend to all other tokens. Therefore, we introduce an additional loss term that encourages the model not only to reconstruct the pixels associated with learned concepts but also to ensure that each token only attends to the image region occupied by the corresponding concept. For instance, as illustrated in Fig. 3, we introduce an attention map regularization term at the position of the entity tokens "*dog*" , "*[cls_0]*", "*cat*" and "*[cls_1]*". Intuitively, the positions within the area containing the entity *e.g.,* "cat", should have larger values than other positions, so we optimize $z_t$ towards the target that the desired area of the object has large values by penalizing the L1 deviation between the attention maps and the corresponding segmentation maps of the entities. We choose $l$ to be the layers with $h_l = w_l = \{32, 16, 8\}$. Formally, we incorporate the following loss terms into the training phase:

$$L_{attn} = \frac{1}{N} \sum_{k=1}^{N} \sum_{l} |CA_l(z_t, y_k) - M_k| \tag{2}$$

where $M_k$ is the segmentation mask of the $k_{th}$ object corresponding to its text token.

**Objective function**. As shown in Fig. 3, given the original clear image $x_0$ and segmented subject image $x_s$, the detected image mask $l_m$ is concatenated to the noisy image latent vector $z_t$ to form a new latent vector $z_t' = concat(z_t, l_m)$. After dimension adjustment through a convolution layer, the feature vector $\tilde{z}_t = conv\_in(z_t')$ is fed into the UNet as the query component. In terms of the conditional information, given the text prompt $y$, $C = T_\theta(v_g, t_y)$ is fused by the text encoder $T_\theta$ from segmented image global embedding ($v_g = I_\theta(x_s)$) and text token embeddings ($t_y$) which are extracted from the fixed CLIP image encoder ($I_\theta$) and the text embedding layer, respectively. For the adapters, they receive local image patch features $v$ and bbox coordinates $l$ as additional information through a MLP feature fusion. Consequently, the Subject-Diffusion training objective is:

$$\mathcal{L} = \mathbb{E}_{\mathcal{E}(x_0), y, \epsilon \sim \mathcal{N}(0,1), t} \big[ \parallel \epsilon - \epsilon_\theta(z_t, t, y, x_s, l, l_m) \parallel_2^2 \big] + \lambda_{attn} L_{attn}. \tag{3}$$

where $\lambda_{attn}$ is a weighting hyper-parameter.

## 4 EXPERIMENTS

### 4.1 IMPLEMENTATION DETAILS AND EVALUATION

The Subject-Diffusion is trained on SDD, as detailed information is provided in Sec. 3.1 and based on SD with implementation details described in Appendix A.

We follow the benchmark *DreamBench* proposed in (Ruiz et al., 2023a) for quantitative and qualitative comparison. In order to further validate the model's generation capability in the open domain, we also utilize the validation and test data from OpenImages, which comprises 296 classes with two different entity images in each class. In comparison, DreamBench only includes 30 classes. We evaluate our method with image alignment and text alignment metrics. For image alignment, we calculate the CLIP visual similarity (CLIP-I) and DINO (Caron et al., 2021) similarity between the generated images and the target concept images. For text alignment, we calculate the CLIP text-image similarity (CLIP-T) between the generated images and given text prompts.

We compare several methods for personalized image generation, including Textual Inversion (Gal et al., 2022), DreamBooth (Ruiz et al., 2023a) and Custom Diffusion (Kumari et al., 2023). All of these models require test-time fine-tuning on personalized images in a certain category. Additionally, we compare ELITE (Wei et al., 2023) and BLIP-Diffusion (Li et al., 2023a), both are trained on OpenImages without test-time fine-tuning.

### 4.2 EXPERIMENTS

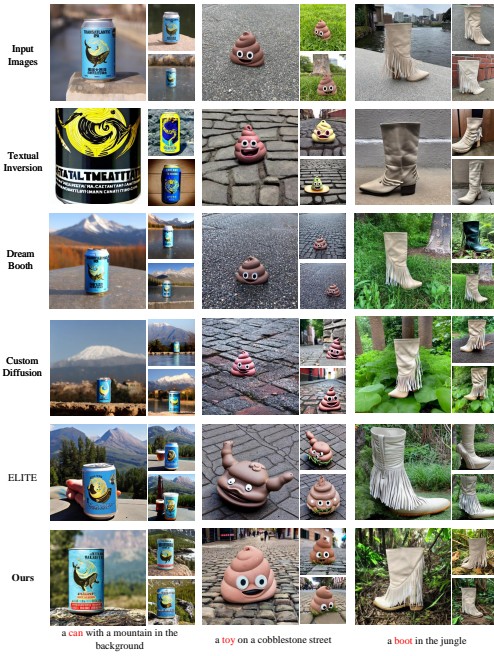

Input Images

Textual Inversion

Dream Booth

Custom Diffusion

ELITE

Ours

a can with a mountain in the background · a toy on a cobblestone street · a boot in the jungle

Figure 4: Qualitative result for single-subject generation. Texture Inversion , DreamBooth and CustomDiffusion employ all three reference images to fine-tune models, whereas only ELITE and Subject-Diffusion can generate personalized images using a single input reference image (corresponding position) without fine-tuning.

Generating personalized images can be a resource-intensive task, with some methods requiring significant storage and computing power to fine-tune models based on user-provided photos. However, our method and similar ones do not require any test-time fine-tuning and can generate personalized images in a zero-shot manner, making them more efficient and user-friendly. In the following sections, we will present both quantitative and qualitative results of our method as compared with other approaches in both single- and multi-subject settings.

**Comparison results for single-subject.** We compare our Subject-Diffusion with the aforementioned 6 methods for single-subject generation. In Table 1, we follow Dreambooth and Blip-diffusion to generate 6 images for each text prompt provided by DreamBench, amounting in total to 4,500 images for all the subjects. We report the average DINO, CLIP-I, and CLIP-T scores over all pairs of real and generated images. The overall results show that our method significantly outperforms other methods in terms of DINO score, with a score of 0.711 compared to DreamBooth's score of 0.668. Our CLIP-I and CLIP-T scores are also slightly higher or on par with other fine-tuning free algorithms, ELITE and BLIP-Diffusion. Furthermore, we conduct experiments on the OpenImages testset, which has about $10\times$ the number of subjects as DreamBench, and our method still achieve high DINO (0.668), CLIP-I (0.782), and CLIP-T (0.303) scores, revealing its generalization ability.

Fig. 4 displays a comparison of the qualitative results of single-subject image generation across various prompts, using different approaches. Excluding Textual Inversion and ELITE, which exhibit significantly lower subject fidelity, our proposed method's subject fidelity and text consistency are comparable to DreamBooth and CustomDiffusion methods that require multiple images for fine-tuning. We have compared our method with Imagen-based methods, including Re-Imagen and SuTI, as shown in Appendix H. Please refer to it for more details.

**Comparison result for multi-subject.**
We conduct a comparison study on our method with two fine-tuning-based approaches, *i.e.,* DreamBooth and Custom Diffusion. This study involves 30 different combinations of two subjects from DreamBench, details of which can be found in Appendix D. For each combination, we generated 6 images per prompt by utilizing 25 text prompts from DreamBench. As depicted in Fig. 5, we present five prompts of generated images. Overall, our method demonstrates superior performance compared to the other two methods, particularly in maintaining subject fidelity in the generated images. On the one hand, images generated by the comparative methods often miss one subject, as exemplified by Dream-Booth's failure to include entities like "on cobblestone street" and "floating on water", as well as Custom Diffusion's inability to accurately capture entities in "on dirty road" and "on cobblestone street". On the other hand, while these methods are capable of generating two subjects, the appearance features between them are noticeably leaking and mixing, leading to lower subject fidelity when

Table 1: Quantitative single subject results. DB denotes DreamBench, and OIT represents the OpenImage testset. † indicates experimental results referenced from BLIP-Diffusion. The value of ELITE is tested by ourself. Boldface indicates the best results of zero shot approaches evaluated in DeramBench. All the comparison methods here are based on the SD model.

| Methods | Type | Testset | DINO | CLIP-I | CLIP-T |
|---|---|---|---|---|---|
| Real Images † | - | - | 0.774 | 0.885 | - |
| Textual Inversion † | FT | DB | 0.569 | 0.780 | 0.255 |
| DreamBooth † | FT | DB | 0.668 | 0.803 | 0.305 |
| Custom Diffusion | FT | DB | 0.643 | 0.790 | 0.305 |
| ELITE | ZS | DB | 0.621 | 0.771 | 0.293 |
| BLIP-Diffusion † | ZS | DB | 0.594 | 0.779 | **0.300** |
| Subject-Diffusion | ZS | DB | **0.711** | **0.787** | 0.293 |
| | | OIT | 0.668 | 0.782 | 0.303 |

compared to the images provided by the user. By contrast, the images generated by our method effectively preserve the user-provided subjects, and each one is accurately produced.

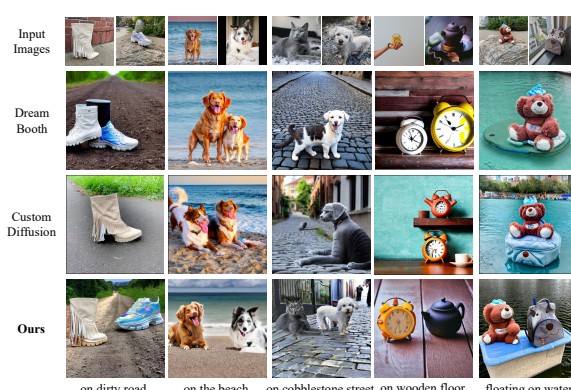

Input Images

Dream Booth

Custom Diffusion

**Ours**

on dirty road    on the beach    on cobblestone street    on wooden floor    floating on water

Figure 5: Qualitative result for multi-subject generation.

We also calculate DINO, CLIP-I and CLIP-T scores on all groups of the generated images, user-provided images and prompts. To obtain CLIP-I, we average the calculated similarities between the generated image and the two subjects, as results presented in Table 2. Obviously, our approach shows remarkable superiority over DreamBooth and Custom Diffusion across DINO and CLIP-T, providing compelling evidence of its ability to capture the subject information of reference images more accurately and display multiple entities in a single image simultaneously.

## 4.3 ABLATION STUDIES

The ablation studies involve examining two main aspects, namely: 1) the impact of our training data and 2) the impact of different components in our Subject-Diffusion model. As shown in Table 3, we present zero-shot evaluation results for both single- and two-subject cases. We observe that all the ablation settings result in weaker quantitative results than our full setting.

Table 2: Quantitative result of two subject generation. ZS means zero-shot and FT denotes fune-tuning. Boldface indicates the best results.

| Methods | Type | DINO | CLIP-I | CLIP-T |
|---|---|---|---|---|
| DreamBooth | FT | 0.430 | 0.695 | 0.308 |
| Custom Diffusion | FT | 0.464 | **0.698** | 0.300 |
| **Subject-Diffusion** | ZS | **0.506** | 0.696 | **0.310** |

**Impact of our training data.** The training data proposed in this paper consists of large-scale, richly annotated images, thereby enabling our model to effectively capture the appearance features of any given subject. To further assess the impact of training data, we retrain our model using OpenImages (Kuznetsova et al., 2020) training data, limiting the categories to only 600. Our evaluation results (a) and (b) demonstrate that this smaller dataset leads to lower image similarity, with the DINO and CLIP-I scores both decreasing for single-subject and two-subject cases, which underscores the importance of utilizing large-scale training data in generating highly personalized images. How-

Table 3: Ablation Results. ↑ and ↓ indicate increase or decrease, respectively. Boldface indicates full-setting results.

| | Index | Methods | DINO | CLIP-I | CLIP-T |
|---|---|---|---|---|---|
| **Single** **Subject** | (a) | **Subject-Diffusion** | **0.711** | **0.787** | **0.293** |
| | (b) | trained on OpenImage | 0.664↓ | 0.777↓ | 0.294↑ |
| | (c) | w/o location control | 0.694↓ | 0.778↓ | 0.275↓ |
| | (d) | w/o box coordinates | 0.732↑ | 0.810↑ | 0.282↓ |
| | (e) | w/o adapter layer | 0.534↓ | 0.731↓ | 0.291↓ |
| | (f) | w/o attention map control | 0.692↓ | 0.789↓ | 0.288↓ |
| | (g) | w/o image cls feature | 0.637↓ | 0.719↓ | 0.299↑ |
| **Two** **Subjects** | (a) | **Subject-Diffusion** | **0.506** | **0.696** | **0.310** |
| | (b) | trained on OpenImage | 0.491↓ | 0.693↓ | 0.302↓ |
| | (c) | w/o location control | 0.477↓ | 0.666↓ | 0.281↓ |
| | (d) | w/o box coordinates | 0.464↓ | 0.687↓ | 0.305↓ |
| | (e) | w/o adapter layer | 0.411↓ | 0.649↓ | 0.307↓ |
| | (f) | w/o attention map control | 0.500↓ | 0.688↓ | 0.302↓ |
| | (g) | w/o image cls feature | 0.457↓ | 0.627↓ | 0.309↓ |

ever, the results still surpass or are on par with those of ELITE and BLIP-diffusion (0.664 vs. 0.621 vs. 0.594 for DINO), demonstrating the effectiveness of Subject-Diffuion's model structure and training strategy.

**Impact of different components.** The comparison between experiments (a) and (c) declares that, if we remove the *location control* (object masks), our model will apparently degenerate over all evaluation metrics. Experiments (a) and (d) indicate that the introduction of *box coordinates* leads to significant improvements in two-subject generation (with the DINO score increasing by 0.042, the CLIP-I score increasing by 0.09, and the CLIP-T score increasing by 0.005). However, the fidelity of single-subject generation decreased by 0.021 for the DINO score and 0.023 for the CLIP-I score. This decline may be due to the fact that, when generating a single subject, the information becomes overly redundant, making it challenging for the model to grasp the key details of the subject.

The high fidelity of our model is primarily attributed to the 256 image patch features input to the adapter layer. As demonstrated in experiment (e), removing this module results in a significant drop in nearly all of the metrics. Experimental results (f) clearly indicate that the *attention map control* operation delivers a substantial performance improvement for two-subject generation as well as a slight performance improvement for single-subject generation. This difference is most likely due to the ability of the attention map control mechanism to prevent confusion between different subjects. The results of (a) and (g) indicate that the absence of the image "CLS" feature led to a significant reduction in the fidelity of the subject, highlighting the significance of the feature in representing the overall image information. Additional qualitative results please refer to the appendix F.

### 4.4    HUMAN IMAGE GENERATION

Due to our method's ability to produce high-fidelity results, it is also well-suited for human image generation. To evaluate our model's effectiveness in this area, we use the single-entity evaluation method employed in FastComposer (Xiao et al., 2023) and compare our model's performance to that of other existing methods. The experimental results are shown in Table 4. Subject-Diffusion significantly outperforms all baseline approaches in identity preservation, with an exceptionally high score that surpasses FastComposer trained on the specific portrait dataset by 0.091. However, in terms of prompt consistency, our method

Table 4: Comparison among our method and baselines on single-subject human image generation. † indicates that the experimental values are referenced from FastComposer.

| Method | Images↓ | ID Preser.↑ | Prompt Consis.↑ |
|---|---|---|---|
| StableDiffusion† | 0 | 0.039 | 0.268 |
| Textual-Inversion† | 5 | 0.293 | 0.219 |
| DreamBooth† | 5 | 0.273 | 0.239 |
| Custom Diffusion† | 5 | 0.434 | 0.233 |
| FastComposer† | 1 | 0.514 | **0.243** |
| **Subject-Diffusion** | **1** | **0.605** | 0.228 |

is slightly weaker than FastComposer (-0.015). We believe this vulnerability could be due to our method's tendency to prioritize subject fidelity when dealing with challenging prompt words.

### 4.5    TEXT-IMAGE INTERPOLATION

By utilizing the "[text prompt], the [subject label] is [PH]" prompt template during image generation, we are able to utilize the dual semantics of both text and image to control the generated image output. Moreover, we could utilize texts and images from distinct categories and perform interpolation of generated images by controlling the proportion of the diffusion steps. To achieve this, we remove the user input image control once the image layout is generated, retaining only the textual semantic control. Our step-based interpolation method is represented by the following formula:

$$\epsilon_t = \begin{cases} \epsilon_\theta(z_t, t, y', x_s, l, l_m) & \text{if } t > \alpha T, \\ \epsilon_\theta(z_t, t, y) & \text{otherwise} \end{cases} \tag{4}$$

In this context, $y$ denotes the original text prompt, while $y'$ signifies employing a convoluted text template: "[text prompt]$_a$, the [subject]$_b$ is [cls]$_a$". The visualization and instructions examples can be found in Appendix E Fig. 8.

## 5    CONCLUSION AND LIMITATION

To date, the high cost and scarcity of manual labeling have posed significant obstacles to the practical implementation of personalized image generation models. Inspired by the breakthroughs in zero-shot large models, this paper develops an automatic data labeling tool to construct a large-scale structured image dataset. Then, we build a unified framework that combines text and image semantics by utilizing different levels of information to maximize subject fidelity and generalization. Our experimental analysis shows that our approach outperforms existing models on the DreamBench data and has the potential to be a stepping stone for improving the performance of personalized image generation models in the open domain. Although our method is capable of zero-shot generation with any reference image in open domains and can handle multi-subject scenarios, it still has certain limitations. First, our method faces challenges in editing attributes and accessories within user-input images, leading to limitations in the scope of the model's applicability. Secondly, when generating personalized images for more than two subjects, our model will fail to render harmonious images with a high probability. Example of failure generations can be found in Appendix G. In the future, we will conduct further research to address these shortcomings.

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

In this supplementary, we will first present the implementation details and training parameters in Appendix A. Then, more details about the dataset construction process and statistics are presented in Appendix B. Then we summarize a comprehensive set of related work comparisons in Appendix C. We further provide more information about the test dataset for two-subject evaluation in Appendix D. We will discuss the interpolation results in Appendix E. We will discuss the Additional qualitative results of the ablation studies in Appendix F. We will discuss the failure cases in Appendix G. We will compare our method with methods trained on the Imagen in Appendix H. And finally, more visualization results of our proposed Subject-Diffusion are exhibited in Appendix I.

## A  IMPLEMENTATION DETAILS

Based on SD v2-base[1], Subject-Diffusion consists of VAE, UNet (with adapter layer), text encoder, and OpenCLIP-ViT-H/14[2] vision encoder, comprising 2.5 billion parameters, out of which a mere 0.7 billion parameters (text encoder, *conv_in* module, adapter layer, and projection matrices $W_K^{(i)}, W_V^{(i)}$) are trainable. The VAE, text encoder, and UNet are initialized from the SD checkpoints, and the CLIP image encoder is loaded from the pretrained OpenCLIP checkpoints. We set the learning rate to 3e-5, the weighting scale hyper-parameter $\lambda_{attn}$ in Eq. (3) to 0.01, and the balance constant $\beta$ in the adapter to 1. The entire model is trained on 24 A100 GPUs for 300,000 steps with a batch size of 12 per GPU. The model is trained based on our proposed SDD or OpenImage training set.

## B  SUBJECT-DIFFUSION DATASET

### B.1  DATASET BUILDING STRATEGY

To produce our dataset, all of our training images are sampled from the LAION-Aesthetics V2 5+[3] which is a subset of LAION-5B with an aesthetic score greater than 5. To keep the diversity of images, we only set the filter conditions for resolution, *i.e.,* keep the images with the small side greater than 1024. However, in order to ensure that the images are suitable for our subject-driven image generation task, we apply several filtering rules: (1) We only keep the bounding boxes with an aspect ratio between 0.3 and 3; (2) We only keep images where the subject's bounding box area is between 0.05 and 0.7 of the total image area; (3) We filter out entities with IOU exceeding 0.8; (4) We remove entities that appear more than 5 times in a detection box; (5) We filter out entities with detection scores below 0.2; (6) We remove images where the segmentation mask area is less than 60% of the corresponding detection box area; (7) For the OpenImages training set, we filter out entities that appear in groups and belong to human body parts. After applying these rules, we keep 22 million images for our SDD and 300,000 images for the OpenImages dataset.

### B.2  STATISTICS AND COMPARISON

Statistics about our training data are illustrated in Fig. 6 and Table. 5. Among them, Fig. 6 presents a comprehensive analysis of the dataset properties of our training data, which includes a detailed distribution of caption length and bbox number per image. The caption length distribution reveals that the majority of captions fall within a range of 5 to 15 words, with a few outliers exceeding 15 words. On the other hand, the bbox number per image distribution shows that most images contain between 1 and 5 bounding boxes, with a small percentage of images having more than 10 bounding boxes. These statistics provide valuable insights into the nature of our training data and can be used to inform the design of our machine learning models.

In Table. 5, we compare the scale of different well-annotated image datasets with the training data used in the study. The number of images in the datasets ranges from 0.028 million to 11 million, while the number of entities ranges from 0.7 million to 1.1 billion. In Table. 5, we compare the scale

---

[1] https://huggingface.co/stabilityai/stable-diffusion-2-base
[2] https://github.com/mlfoundations/open_clip
[3] https://huggingface.co/datasets/ChristophSchuhmann/improved_aesthetics_5plus

Table 5: The comparison between well annotated image dataset and our training data. Image # , entity # and class # refer to the number of images, the number of entities and the number of class categories, respectively. SA-1B † does not provide the class label of instances.

| Dataset | LVIS v1 | COCO | ADE20K | Open Images | SA-1B † | SDD (ours) |
|---------|---------|------|--------|-------------|---------|------------|
| Image # | 0.120M | 0.123M | 0.028M | 1M | 11M | 76M |
| Entity # | 1.5M | 0.9M | 0.7M | 2.7M | 1.1B | 222M |
| Class # | 1200 | 91 | 2693 | 600 | N/A | 162K |

of different annotated image datasets to the training data used in our study. The number of images in these datasets ranges from 28,000 to 11 million, with the entity count ranging from 700,000 to 1.1 billion. Although SA-1B (Kirillov et al., 2023) offers the highest entity count of 1.1 billion, it lacks annotated entity categories and tends to include small-sized masks, which is unsuitable for our image generation purposes. In contrast, the training dataset employed in this study comprises 76 million images and 220 million entities, making it the largest-scale dataset available. Furthermore, it is important to note that our study not only provides the number of entity classes but also highlights the superior diversity of our training data compared to other datasets. This diversity is crucial in enabling our model to comprehend and identify a wide range of reference objects in the open world. Our training data includes a vast array of entities, *i.e.* 162K kinds of entities, ranging from common objects such as animals and plants to more complex entities such as vehicles and buildings. This comprehensive dataset ensures that our model is equipped with the necessary knowledge to accurately identify and classify any reference object it encounters. Additionally, our study also takes into account the varying contexts in which these entities may appear, further enhancing the robustness and adaptability of our model. Overall, our research provides a comprehensive and diverse training dataset that enables our model to effectively understand and generate reference objects in the open world.

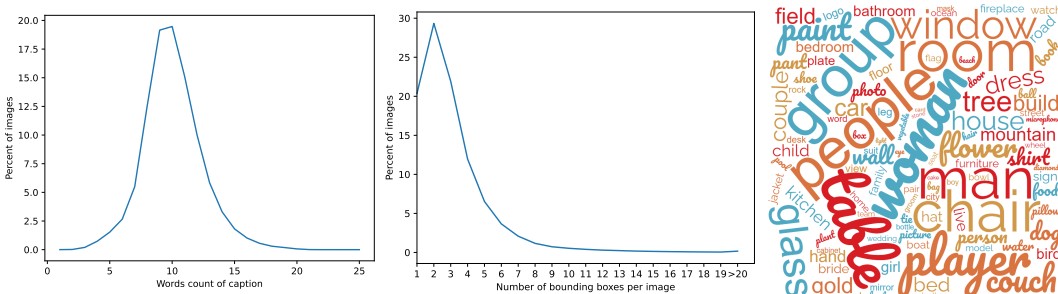

Figure 6: Dataset properties. Left: word count distribution of captions in SDD; Middle: bounding box count distribution of images in SSD; Right: Word cloud diagram of SDD. We can observe that the most frequent entities in our SDD are man, woman, people, table, room, *etc.*

## B.3 DISCUSSION ON QUALITY OF THE DATA

We collected 1000 data samples for statistics, and some of the figures are presented in Fig. 7. We also conducted an analysis of four columns of sample data, where the first three columns on the left are the data we selected for training after rule-based filtering, and the column on the right represents the data excluded by the filtering rules. The first column on the left shows high-quality data selected subjectively by the annotators, with filtering criteria consistent with our rule-based filtering motivations. The second column on the left shows low-quality results with low recall, i.e., many subject entities are not detected by the bounding box, possibly due to the generation of corresponding entities being incomplete in BLIP2 or insufficient recall by DINO. The third column on the left corresponds to other low-quality situations, which may include errors in subject identification, i.e., low accuracy, or situations missed by the rule-based filtering. Finally, we conducted a simple analysis of 1000 samples, as shown in Table 6, Subjectively high-quality results only accounted for 35% of

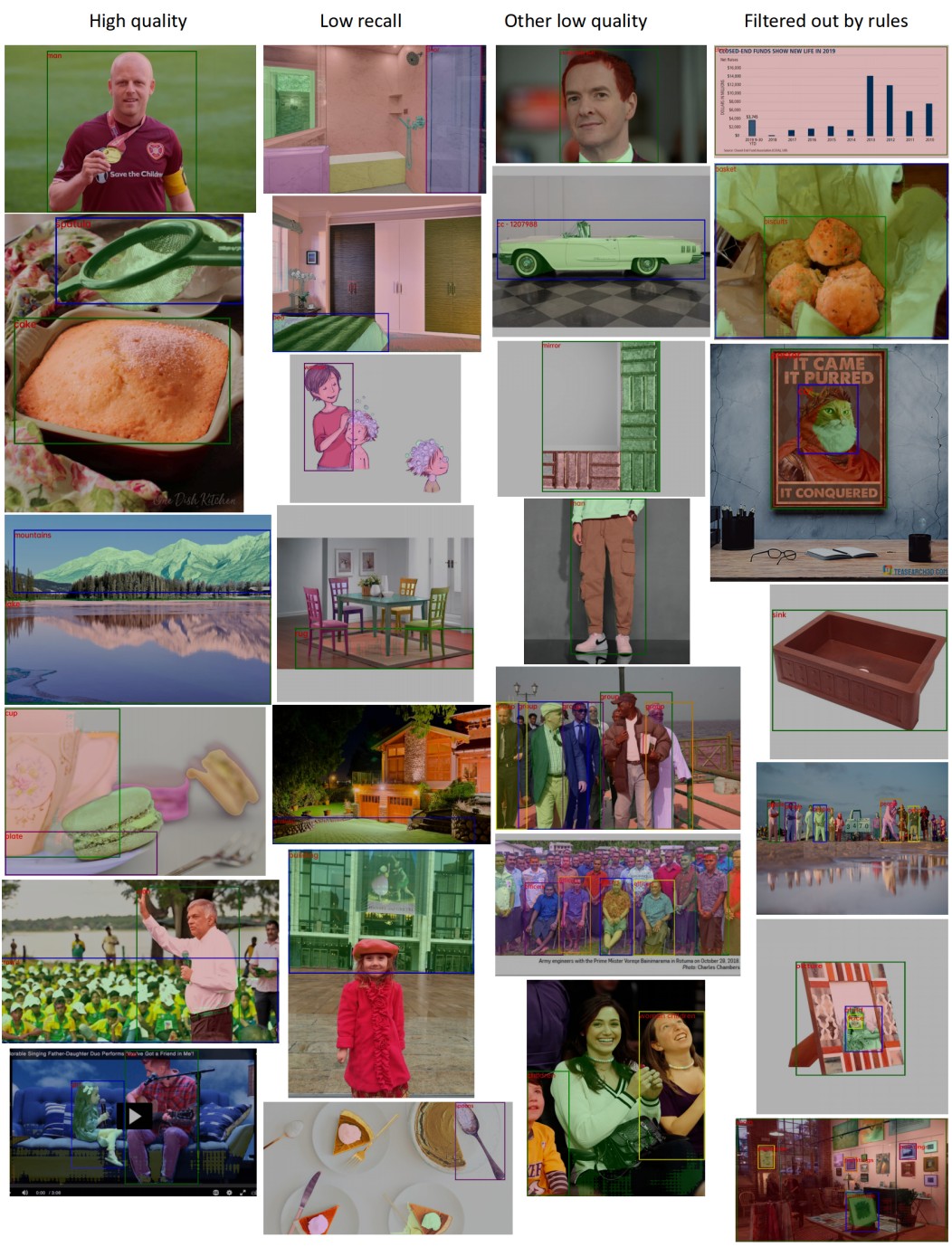

Figure 7: Example of data of different qualities.

the rule-based filtering results. This indicates that there is still a lot of potential to optimize data quality, and we will continue to work hard in this area.

## C  PERSONALIZATION BASELINES COMPARISON

We carefully survey the personalized image generation papers published in recent years and compile a comprehensive comparison table comparing their support for single reference image, multi-subject

Table 6: Subjective quantitative statistics of data quality.

| Data Quality | High Quality | Low Recall | Other Low Quality | Filtered Out by Rules |
|---|---|---|---|---|
| Proportion | 18% | 20% | 14% | 48% |

generation, no test-time fine-tuning, and open domain generalization. As delineated in Table 7, the main stream of personalized image generation still considers test-time fine-tuning, which suffers from inference time-consuming ranging from several seconds to more than one hour (Gal et al., 2022; Ruiz et al., 2023a; Kumari et al., 2023; Gal et al., 2023; Han et al., 2023b; Smith et al., 2023; Voynov et al., 2023; Liu et al., 2023b;c; Tewel et al., 2023; Chen et al., 2023a; Avrahami et al., 2023; Alaluf et al., 2023; Gu et al., 2023; Hao et al., 2023; Ruiz et al., 2023b; Arar et al., 2023; Zhou et al., 2023). Only a small portion of papers are dedicated to studying personalized image generation without test-time fine-tuning (Jia et al., 2023; Shi et al., 2023; Xiao et al., 2023; Chen et al., 2023c; 2022; Ma et al., 2023b; Wei et al., 2023; Li et al., 2023a; Chen et al., 2023b). But all of the pioneering works cannot satisfy the four aforementioned requirements, either by being trained on specific domains (Shi et al., 2023; Jia et al., 2023; Xiao et al., 2023), or by supporting only single-concept generation. To the best of our knowledge, our Subject-Diffusion is the first open-domain personalized image generation method that supports multi-concept synthesis and requires only a single reference image for each subject.

Table 7: Survey of recent personalized image generation works in terms of single reference image, multi-subject generation, no test-time fine-tuning and open domain generalization.

| Method | Single image | Multi-subject | No fine-tuning | Open domain |
|---|---|---|---|---|
| Textual Inversion (Gal et al., 2022) | ✗ | ✗ | ✗ | - |
| Dreambooth (Ruiz et al., 2023a) | ✗ | ✗ | ✗ | - |
| Custom Diffusion (Kumari et al., 2023) | ✗ | ✓ | ✗ | - |
| E4T (Gal et al., 2023) | ✓ | ✗ | ✗ | - |
| SVDiff (Han et al., 2023b) | ✓ | ✓ | ✗ | - |
| Continual Diffusion (Smith et al., 2023) | ✗ | ✓ | ✗ | - |
| XTI (Voynov et al., 2023) | ✗ | ✗ | ✗ | - |
| Cones (Liu et al., 2023b) | ✓ | ✓ | ✗ | - |
| Cones 2 (Liu et al., 2023c) | ✓ | ✓ | ✗ | - |
| Perfusion (Tewel et al., 2023) | ✗ | ✓ | ✗ | - |
| DisenBooth (Chen et al., 2023a) | ✓ | ✗ | ✗ | - |
| Break-A-Scene (Avrahami et al., 2023) | ✓ | ✓ | ✗ | - |
| NeTI (Alaluf et al., 2023) | ✗ | ✗ | ✗ | - |
| Mix-of-Show (Gu et al., 2023) | ✗ | ✓ | ✗ | - |
| ViCo (Hao et al., 2023) | ✗ | ✗ | ✗ | - |
| HyperDreamBooth (Ruiz et al., 2023b) | ✓ | ✗ | ✗ | - |
| Domain-Agnostic (Arar et al., 2023) | ✓ | ✗ | ✗ | - |
| Regularization-Free (Zhou et al., 2023) | ✓ | ✗ | ✗ | - |
| Taming (Jia et al., 2023) | ✓ | ✗ | ✓ | ✗ |
| InstantBooth (Shi et al., 2023) | ✓ | ✗ | ✓ | ✗ |
| PhotoVerse (Chen et al., 2023b) | ✓ | ✗ | ✓ | ✗ |
| Face0 (Valevski et al., 2023) | ✓ | ✗ | ✓ | ✗ |
| FastComposer (Xiao et al., 2023) | ✓ | ✓ | ✓ | ✗ |
| SuTI (Chen et al., 2023c) | ✗ | ✗ | ✓ | ✓ |
| Re-Imagen (Chen et al., 2022) | ✓ | ✗ | ✓ | ✓ |
| UMM-Diffusion (Ma et al., 2023b) | ✓ | ✗ | ✓ | ✓ |
| ELITE (Wei et al., 2023) | ✓ | ✗ | ✓ | ✓ |
| Blip-Diffusion (Li et al., 2023a) | ✓ | ✗ | ✓ | ✓ |
| Ours (Subject-Diffusion) | ✓ | ✓ | ✓ | ✓ |

## D   TWO-SUBJECT EVALUATION DETAILS

We utilize all the objects in DreamBench and randomly select 30 pairs of combinations, out of which 9 pairs belong to live objects. The specific subject pairs are presented in Table 8. For the prompts used in generating images with two subjects, we follow the format outlined in DreamBench, with the two subjects connected using the word "and".

For inference, we use PNDM scheduler for 50 denoising steps. We use a fixed text guidance scale 3 and image guidance scale 1.5 for all experiments

Table 8: Prompts for a dual-subject personalized image generation testset. The first 21 combinations are still objects, and the last 9 combinations are animals.

| backpack-can | bear_plushie-backpack_dog | berry_bowl-vase |
|---|---|---|
| duck_toy-can | fancy_boot-shiny_sneaker | grey_sloth_plushie-poop_emoji |
| teapot-backpack_dog | teapot-berry_bowl | wolf_plushie-backpack_dog |
| can-bear_plushie | can-candle | can-duck_toy |
| can-shiny_sneaker | clock-teapot | colorful_sneaker-vase |
| robot_toy-backpack | shiny_sneaker-duck_toy | shiny_sneaker-poop_emoji |
| pink_sunglasses-candle | poop_emoji-clock | poop_emoji-shiny_sneaker |
| cat-dog2 | cat-dog5 | cat2-dog3 |
| dog2-dog3 | dog5-dog6 | dog6-dog7 |
| dog6-dog8 | dog7-dog8 | dog8-dog6 |

## E   TEXT-IMAGE INTERPOLATION

The visualization examples can be found in Fig. 8. We provide this experiment to show that the high-level information of the user-provided images are successfully extracted and rendered in generated images during early backward diffusion stages. Thus we can adjust $\alpha$ to balance image fidelity and editablity according to different prompts.

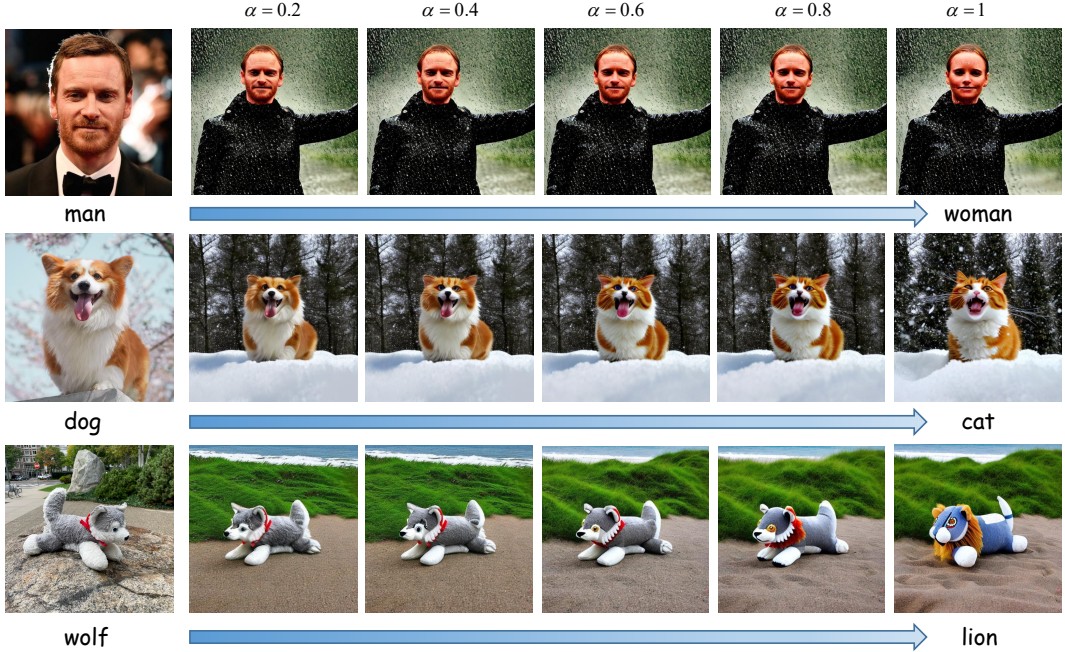

Figure 8: Text-image interpolation. The prompts are followings: *A man in the rain, the woman is [PH]*; *A dog in the snow, the cat is [PH]*; *A wolf plushie on the beach, the lion is [PH]*.

## F   ADDITIONAL QUALITATIVE RESULTS OF THE ABLATION STUDIES

In the case of a single subject, Fig. 10 left two columns present two examples that clearly demonstrate the higher fidelity of the generated images without box coordinates. However, these images have lower semantic matching ability and are unable to capture key information from the prompts. On the other hand, images generated without the adapter layer and without image cls feature have slightly lower fidelity. These two strategies aim to enhance the processing of input image information, providing advantages in both objective metrics and subjective evaluation in terms of fidelity.

Regarding the case of two subjects with Fig. 10 right two columns, the conclusions remain consistent with the previous analysis. Images generated without the adapter layer and without image cls feature still exhibit slightly lower fidelity. It is worth mentioning that both the preservation of box coordinates and attention map control have advantages in generating images with multiple subjects, as these conditions help alleviate the issue of generating ambiguous representations of multiple entities.

(a) Failing to edite attributes

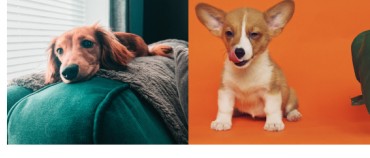

Input Image          A black haired dog          Input Image          A crying man playing ball
                     running  on the water

(b) Failing to render harmonious images with two subjects

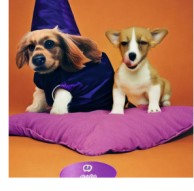 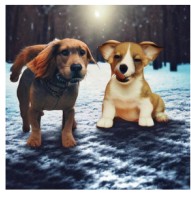

Input Image          A dog and a dog in a          A dog and a dog in the snow
                     purple wizard outfit

Figure 9: Example of failure generations.

## G   FAILURE CASES DISCUSSIONS

We provide an example to address the shortcomings of "editing attributes" and "rendering harmonious images with two subjects". For the "editing attributes" issue, the attributes corresponding to the red-marked prompts in the failed image are highlighted. As for generating images with two subjects, if the source image(s) itself already lacks one or both of the subjects, it may lead to disharmony in the final generated image.The cases are shown in Fig. 9.

## H   DISCUSSIONS WITH METHODS TRAINED ON THE IMAGEN

From Table 9, We have compared our method with Imagen-based methods, including Re-Imagen and SuTI. Re-Imagen is a retrieval-augmented approach that also achieves personalized image (retrieved reference image) generation. SuTI is a subject-driven text-to-image generator that replaces subject-specific fine tuning with in-context learning. We can see that SuTI has an advantage in all three metrics. However, it may not be fair to make direct comparisons between the two methods based

Figure 10: Additional qualitative results of the ablation studies.

solely on these results.Three issues that need to be discussed are as follows: First, the difference in the base model used, where SuTI is based on the Imagen model structure and Initialization parameters, while our base model is SD2. Second, the image resolution evaluated for SuTI was 1024, while our evaluated image resolution was 512. Third, SuTI provides four demonstration image-text pairs during inference, while we only provide one.

we will compare our results with SuTI in a qualitative side-by-side comparison in Figure 11. We made a simple comparison on the four shortcomings of SuTI:

(1) SuTI has a strong prior about the subject and hallucinates the visual details based on its prior knowledge. For example, the generation model believes 'teapot' should contain a 'lift handle'. (2) Some artifacts from the demonstration images are being transferred to the generated images like second column.Subject Diffusion has advantages in this regard because it removes background input. (3) The subject's visual appearance is being modified through with SuTI, mostly influenced by the context, like last column. Subject Diffusion will be slightly better. (4) SuTI is not particularly good at handling compositional prompts like the 'sunglasses' example like third column. Subject Diffusion will be slightly better.

Table 9: Quantitative single subject results. † indicates experimental results referenced from SuTI. Boldface indicates the best results of zero shot approaches evaluated in DeramBench.

| Methods | Model Base | Testset | DINO | CLIP-I | CLIP-T |
|---|---|---|---|---|---|
| Real Images † | - | - | 0.774 | 0.885 | - |
| Re-Imagen † | Imagen | DB | 0.600 | 0.740 | 0.270 |
| SuTI † | Imagen | DB | **0.741** | **0.819** | **0.304** |
| **Subject-Diffusion** | SD | DB | 0.711 | 0.787 | 0.293 |

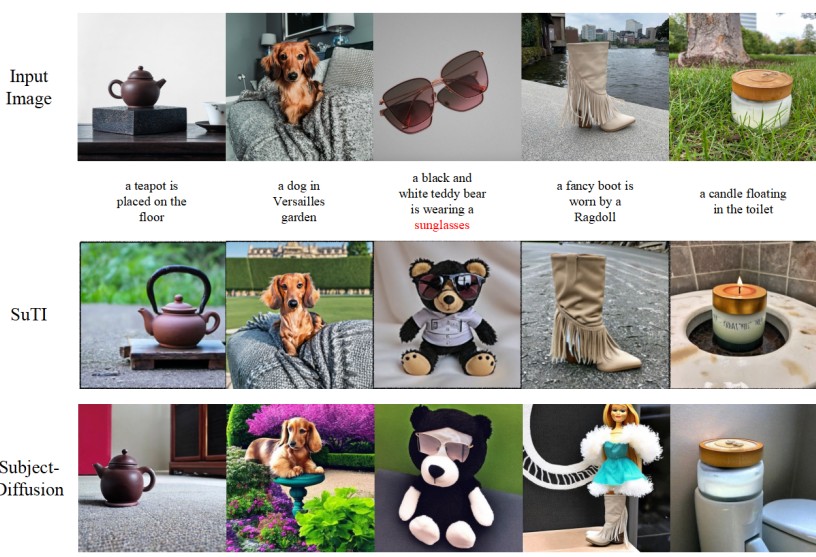

Figure 11: Compare our results with SuTI in qualitative.

# I MORE VISUALIZATION RESULTS

In this section, we provide more single-, multi-, and human subject generation visualization examples, as in Fig. 12, Fig. 13 and Fig. 14. Notice that we display 10 generated results for each personal image without carefully cherry-picking, demonstrating the consistent fidelity and generalization ability of our proposed Subject-Diffusion.

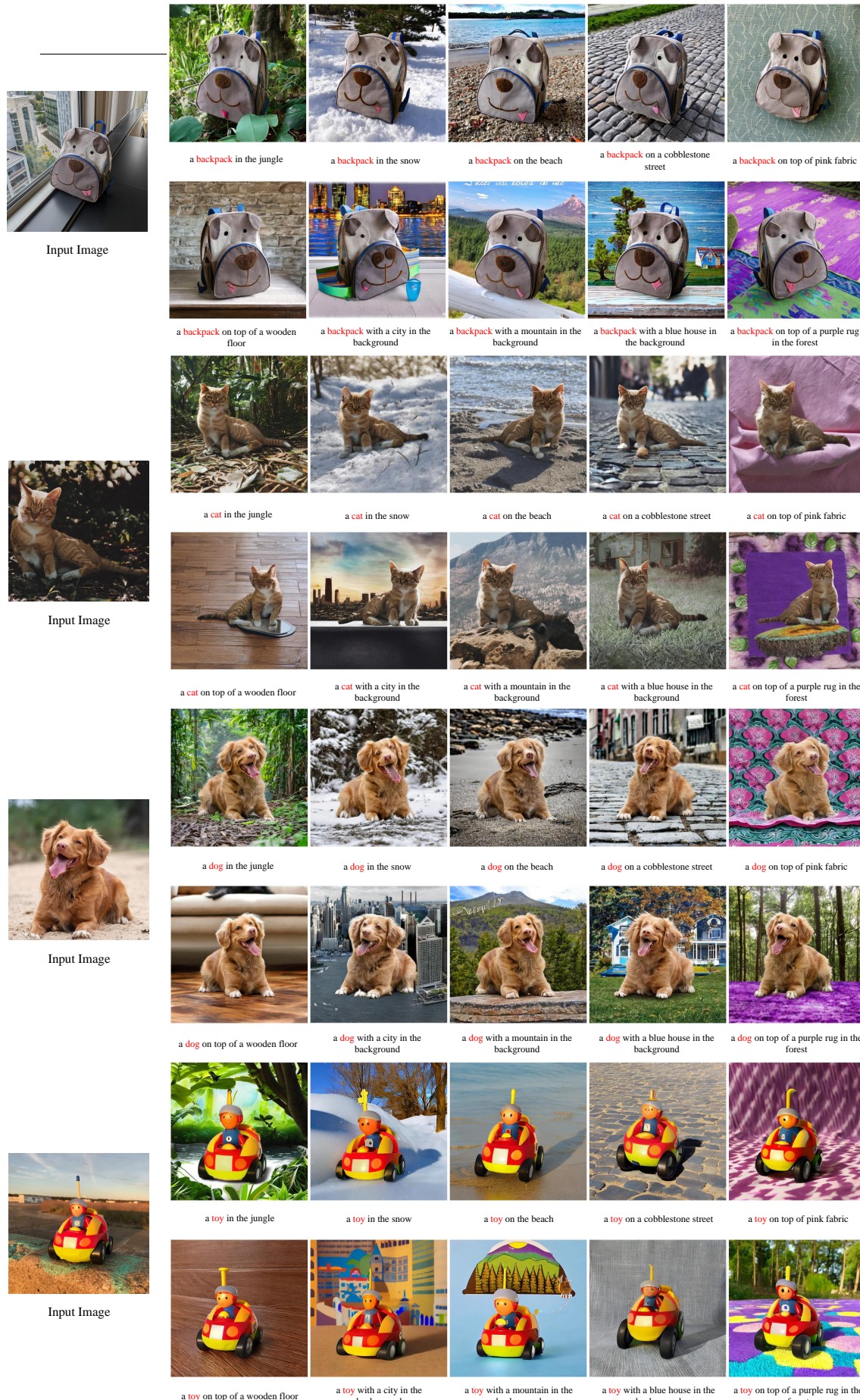

Figure 12: More qualitative results for single-subject generation.

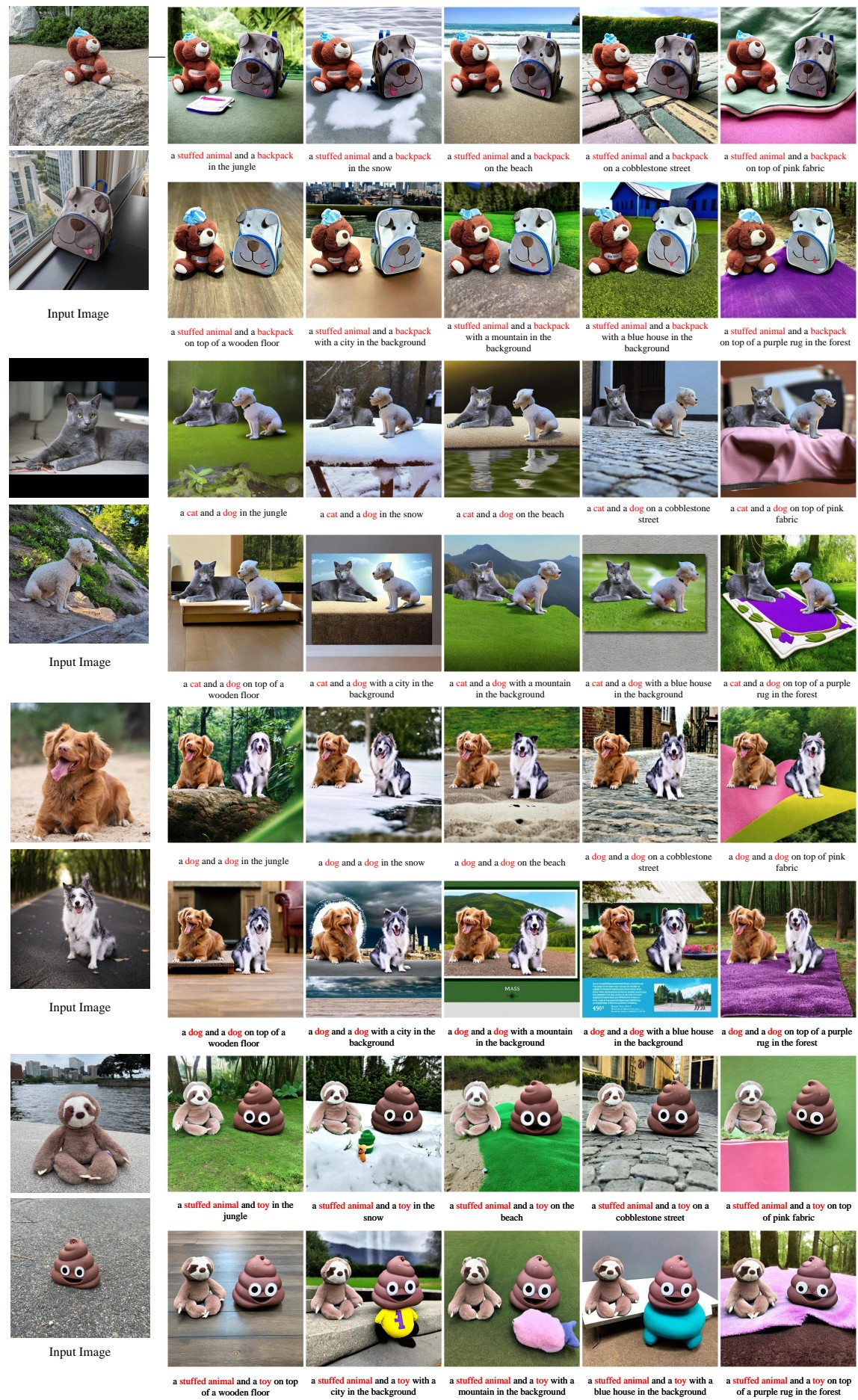

Figure 13: More qualitative results for two-subject generation.

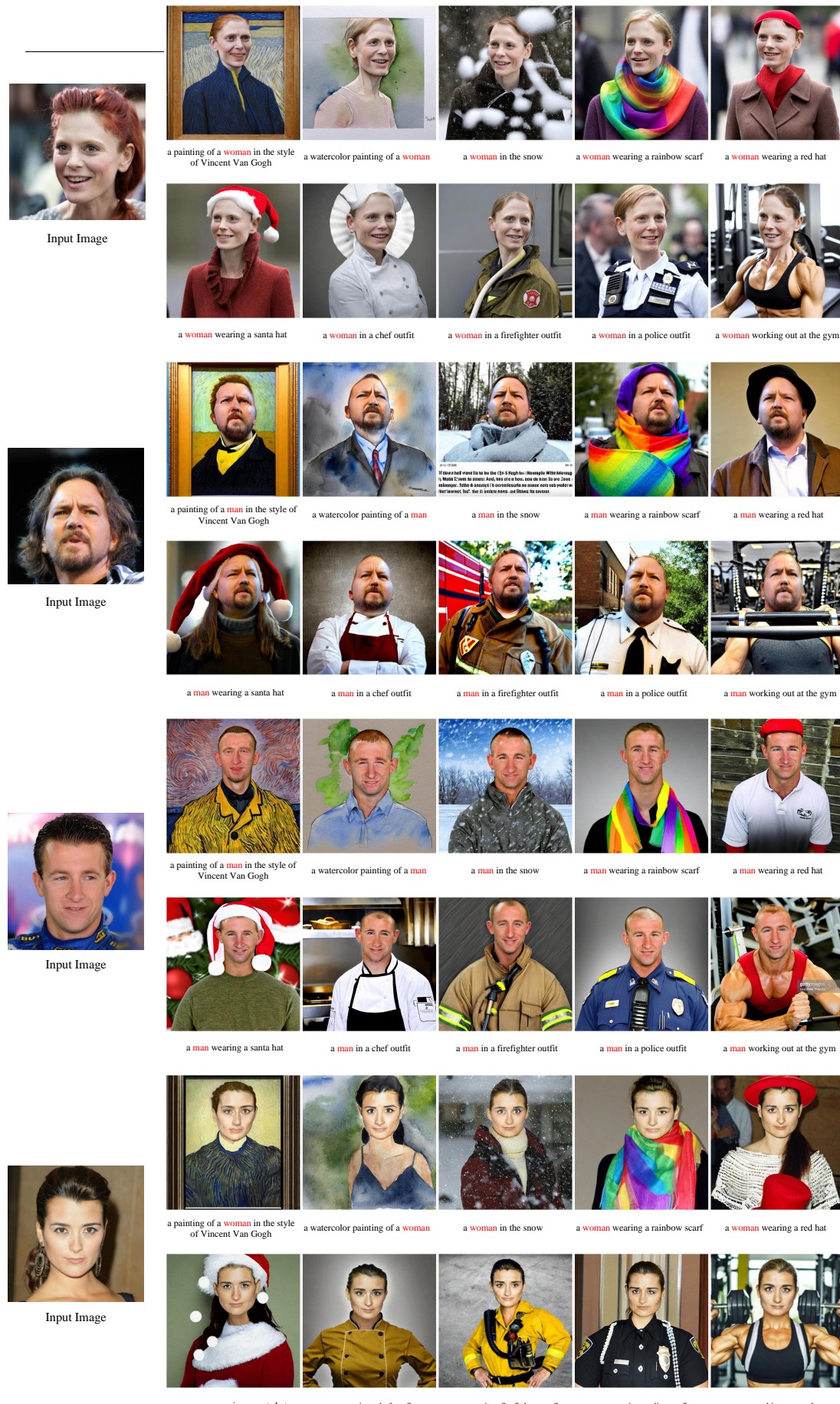

Figure 14: More qualitative results for human image generation.

