# OpenReview forum: "Subject-Diffusion: Open Domain Personalized Text-to-Image  Generation without Test-time Fine-tuning"
_ICLR.cc/2024/Conference — Submitted to ICLR 2024_

### Official Review · Reviewer_8qrS · 2023-10-29

**Soundness:** 2 fair
**Presentation:** 2 fair
**Contribution:** 2 fair
**Rating:** 3
**Confidence:** 4

**Summary:**

This paper proposes to learn subject-driven image generation from training adapter layers for diffusion model, on a large-scale subject-related image generation dataset. Moreover, this paper designs the model architecture to take control signals such as bounding box, segmentation mask to supplement its model training for fidelity and faithfulness. Using this model, the paper claims that it can perform single subject image generation, multi-subject image generation, as well as human subject generation.

**Strengths:**

1. The recipe of adapting pre-trained diffusion model into subject-driven image generation is novel and interesting.
2. The trained model is capable of doing multiple subject-related text-to-image generation tasks, via careful prompting the model, which is nice.
3. The quantitative results look strong to all the methods the authors have compared with.
4. The ablation study looks comprehensive.

**Weaknesses:**

- The paper is emphasizing that they have outperformed state-of-the-art in single subject generation, which is not supported by its results. Particularly, the comparison in Table 1 is intentionally ignoring the state-of-the-art results from SuTI (Chen et. al. 2023). I have found neither comparison nor discussion to justify this ignorance. Additionally, it would also be important to do qualitative side-by-side comparison with SuTI to understand the quality of generation.
  - IMO, a great paper could be one without the state-of-the-art performance, but should not be one that claims to be state-of-the-art without comparing to the actual state-of-the-art method.
- The reproducibility of the paper is questionable, given that the model training is relying on a large private dataset (at least the labels, masks, and grounding). While the dataset is being argued as one of this paper's main contribution, there is no discussion on quality of the data (not a lot examples in main paper or appendix), no discussion on how bias/privacy is protected, no plan or discussion on releasing the data.
- The image resolution provided in the paper is quite small (even counting the ones in the supplementary). Having high-resolution image generation is quite important to assess the quality of model in the field of image generation. Looking at figure 6 (when zooming in ), I could find a lot of artifacts in the interpolated results, the face is losing details, the cat face is weird, the lion looks more like a horse to me. Similarly, figure 5 also seems to have a lot of artifacts from my perspective.

**Questions:**

- During the inference on single-subject / multi-subject generation, do you provide those auxiliary information to the model?
  - If the mask  / segmentations are available for the subject-diffusion only during the inference, what would be the performance if they are removed during the inference? Also how faithful is the model's generation to those control signal?
  - the mask  / segmentations are not available in inference, would the model's performance improve given those controls signal?

**Details Of Ethics Concerns:**

The paper presents a diffusion model specific to subject (particularly human subject), which was trained on a large web crawl of images (that collects human faces), without any particular procedure to ensure the privacy of the personal data, as we as performing any responsible AI processing. It is likely that the model trained on such data is under-representing certain group than the others. However, there is no analysis on the distribution of data (in terms of ethic groups), no privacy filtering (e.g., anonymizing/mosaicing human face, or at least de-associate the name of the person with the face), etc.

Given that a big part of the contribution from this paper is the dataset, I believe such analysis and some degree of anonymization is required, and would like to ask the ethic reviewer to consider reviewing this particular paper.

---

> ### Author Response · Authors · 2023-11-20
>
> W1-A1: Firstly, we would like to sincerely apologize for our negligence in not thoroughly reviewing and organizing the experimental results related to the topic. This led to the omission of an important comparison result. We want to emphasize that we did not intentionally ignore this result (as we have cited this paper in Appendix C "Personalization Baselines Comparison"). As researchers, we would never intentionally violate principles. However, due to our negligence, we missed its evaluation in DreamBench.
>
> Over the past few days, we have analyzed the results of the SuTI paper carefully. It must be admitted that with DINO, CLIP-I, and CLIP-T scores, subject-diffusion is all lower than those of SuTI on the same evaluation set.
> Then the three issues that need to be discussed are as follows:
> First, the difference in the base model used is that SuTI is based on the Imagen model structure and initialization parameters, while our base model is SD2.
> Second, the image resolution evaluated for SuTI was 1024, while our evaluated image resolution was 512.
> Third, SuTI provides four demonstration image-text pairs during inference, while we only provide one.
> Due to time constraints for the rebuttal, it is difficult for us to fairly compare our methods based on the Imagen architecture and initialization parameters.
> Of course, we remove the description of "outperformed state-of-the-art" in our paper and remove the Re-Imagen experiment results from Table 1. We only include the results based on the SD model as a comparison. We also provide supplementary explanations for the Re-Imagen and SuTI trained on the Imagen base in the experiments section, and we also include the metrics in Appendix H for a simple comparison. Additionally, we compared our results with SuTI in a qualitative side-by-side comparison in Appendix H marked in red.  Thank you for your suggestion. We analyze from qualitative indicators and compared the four limitations mentioned in the SuTI article. Subject diffusion has certain advantages.
> Once again, we apologize for our negligence.
>
>
> W2-A2: We have been releasing the data all along, but due to the strict open source process within the company, the period is long. However, ultimately, it is feasible for us to move towards open source. The entire dataset is almost 40T in size, so we are currently organizing our open source content. As the original images themselves from LAION are open source, we plan to organize the URLs of the original images and then open source the corresponding segmented images that we have processed along with structured information such as BLIP captions, masks, and boxes.
> In Appendix B.3 marked in red, we provide additional subjective quantitative statistics on data quality for reference.
> Although we have not yet released the data, we have open-sourced our code. We are aware of one work, "High-fidelity Person-centric Subject-to-Image Synthesis," that has reproduced our paper and compared it with our method.
>
>
> W3-A3: The resolution of the generated images on paper is 512.  This is likely because the quality of the generated images is mainly determined by the parameters of the UNet model. For this part, we only trained the adapter and cross-attention K and V matrix parameters, so we speculate that this is probably an inherent issue with the base model SD2.
> Indeed, it is difficult to compare the base models without providing a metric such as FID or other measures to evaluate the generated images, as Subject-Diffusion does not support being used as a regular T2I model without additional reference image or other input conditions.We will continue to monitor this issue in the future.
>
>
> Q-A: During inference, the mask plays two roles in both single-subject and multi-subject generations.
> The first role is to control the specific location of the subject to be generated in the final image. In our business application, this input can be randomly set, but with some degree of adherence to rules to control its freedom. For example, we may adjust the mask size and position based on the user's input reference image to better control the degree of freedom and meet the user's actual needs.
> The second role is to enhance the fidelity of the subject. Without the mask, the subject can still be generated, but the fidelity will be significantly decreased. According to our previous statistical results, the DINO metric reduces to a level similar to that of Textual Inversion.
>
> Segmentation also plays the same role in both single-subject and multi-subject generations during inference. If the segmentation is not available, the generated results will be more heavily influenced by the background of the reference image. This is why we used SAM to remove the background effect during training. Both of these control conditions are consistent with the training process.
>
> We are looking forward to your reply and happy to answer your further questions.
>
> Best regards
>
> Authors

---

### Official Review · Reviewer_4YXM · 2023-10-31

**Soundness:** 3 good
**Presentation:** 3 good
**Contribution:** 3 good
**Rating:** 6
**Confidence:** 5

**Summary:**

The paper introduces "Subject-Diffusion," a novel approach for open-domain personalized image generation that doesn't require test-time fine-tuning. The model only needs a single reference image to support personalized generation of single or multiple subjects in any domain. The authors constructed a large-scale dataset with 76M images and their corresponding subject detection bounding boxes, segmentation masks, and text descriptions. The proposed framework combines text and image semantics, incorporating location and fine-grained reference image control to maximize subject fidelity and generalization. The results indicate that the method outperforms other state-of-the-art frameworks in various image generation tasks.

**Strengths:**

- Introduces a new approach that doesn't require test-time fine-tuning, addressing a significant challenge in the field.
- Utilizes a single reference image, making it more user-friendly and versatile.
- Incorporates a comprehensive dataset with 76M images, enhancing the model's training and performance.
- Combines text and image semantics, leading to high fidelity and generalization in generated images.
- Demonstrates superior performance compared to other state-of-the-art methods when generating multiple objects

**Weaknesses:**

- The construction of the large-scale dataset might have biases or inconsistencies that could affect the model's performance.

**Questions:**

None

---

> ### Author Response · Authors · 2023-11-17
>
> Thank you for your review. You are correct that the quality of the data can have a significant impact on the final results. We initially trained our models using the OpenImages dataset, we found that the performance metrics were lower  (but still outperforms ELITE and Blip-diffusion as shown in Table 1 and Table 3 ), especially for less common themes, such as certain artworks. This inspired us to construct large-scale structured data as thoroughly as possible to achieve better learning results. We used the largest open-source image-text matching dataset LAION, to ensure that as much learnable knowledge was included as possible.
> Additionally, different screening criteria for the structured data, such as the shape, size, position, and number of entities in the bounding boxes and segmented images, can have a significant impact on the results. Our final screening principle was to maximize the quality of the data while ensuring that the data volume reached a certain scale. Of course, this required a significant amount of time and effort.
> Finally, we are also currently working on organizing the data and making it available to the public in an open format to contribute to the development of personalized generation technology.

---

> > ### Comment · Reviewer_4YXM · 2023-11-21
> > **Thanks for response**
> >
> > Thank you for your response.
> >
> > After reviewing your rebuttal and other reviewers' concerns, I would like to keep my score.

---

### Official Review · Reviewer_Gk6j · 2023-10-31

**Soundness:** 2 fair
**Presentation:** 2 fair
**Contribution:** 2 fair
**Rating:** 5
**Confidence:** 4

**Summary:**

This paper proposes Subject-Diffusion, an open-domain personalized text-to-image generation model that can support single- or multi-subject personalization using a single reference image without test-time model fine-tuning. The authors develop an automatic data labeling tool and construct a large-scale dataset that comprises 76M open-domain images and 222M entities. They introduce a unified framework that combines text and image semantics by incorporating coarse location and fine-grained reference image control. The authors design the prompt format and employ a trainable text encoder, as well as insert an adapter between each self- and cross-attention block, encoding dense patch features of the segmented objects and their corresponding bounding box information. The framework also adopts an attention control mechanism to support multi-subject generation. Experiments demonstrate the advantages of the proposed method over state-of-the-art baselines.

**Strengths:**

- The paper is generally well-written. The symbols, terms, and concepts are adequately defined.

- Sufficient details are provided to explain the proposed method. The framework shows some advantages over existing baselines.

- The relevant literature is well-discussed and organized.

**Weaknesses:**

- The reviewer's primary concern is the actual experimental performance. In many results generated by Subject-Diffusion (e.g., Figure 1), the subjects lack diversity/identity variation. This suggests that Subject-Diffusion may have limited creative generation capability and generalizability. Some results are akin to image composition.

- Apart from the quantitative results, presenting additional qualitative results of the ablation studies would strengthen this paper further.

- The reviewer is interested in the accuracy of the automatic data labeling tool. Providing more evaluation and analysis of the tool to demonstrate its merits is beneficial.

- The presentation of this paper could be improved. Most images in the figures of the paper are too small, which requires careful zooming in to check details. Besides, the layout of some sections, especially the experiments, is a bit messy.

**Questions:**

- About the proposed framework, what is the intuition of fixing the image encoder while training the entire text encoder?

- Will the authors release the proposed SDD dataset? It is very useful if the dataset will be made publicly available. Also, the code should be made publicly available to ensure reproducibility.

- The limitation and failure case discussions are missing, which are highly recommended to be included.

- The last line of Page 1 has an Appendix reference error. This also applied to other places referring to the Appendix.

- Regarding the quantitative results in Table 1, some results are borrowed from BLIP-Diffusion while others are tested. It would be useful if the authors could provide more details on the test settings to ensure a fair comparison.

---

> ### Author Response · Authors · 2023-11-17
>
> W1-A1: It is true that our method lacks diversity/identity variation, and although we have made great efforts to balance diversity and fidelity in our strategy design, we have not yet achieved sufficient improvement in diversity. This is mainly due to the fact that during training, the individual samples and corresponding structured information are independent of each other. To improve diversity, we rely mainly on data augmentation techniques applied to the segmentation image, but this method is still weak. A more reasonable approach would be to retrieve and match segmented images from a database, then calculate the similarity between them to obtain condition inputs with certain attribute variations. We will actively try to implement such diversity-enhancing strategies in the future.
>
> Furthermore, the use of a single-image input for inference without requiring finetuning has its limitations. Similar issues can be observed in models like FastComposer and SuTI that use multiple images without requiring finetuning. Balancing personalized image generation with diversity and fidelity still has a long way to go, and we will continue to work towards better solutions.
>
> W2-A2: We provide additional qualitative results of the relationship studies, please refer to our Appendix F marked in red.
>
> W3-A3: We provide subjective quantitative statistics of data quality, please refer to our Appendix B.3 marked in red.
>
> W4-A4: Sorry, we overlooked this. We have corrected it. Thanks for the reminder.
>
>
>
> Q1-A1: The input source of the image encoder for training the subject of the image is where the feature extractor comes into play, extracting relevant features. The motivation for training the entire text encoder is more direct, as its input introduces features of the image modality through certain rules of prompts. In order to better incorporate features of both modalities, it is necessary to shuffle the text encoder. However, we believe that this intuition requires more powerful experimental verification. In fact, we have verified that training only the last layer or two layers of blocks of the image encoder, or training only the first two layers of the text encoder, results in slightly worse actual experimental performance. However, because we did not think these ablation studies were the focus of the paper, we did not include them in the article.
>
> Q2-A2: We have been implementing this plan all along, but due to the strict open source process within the company, the period is long. However, ultimately, it is feasible for us to move towards open source. The entire dataset is almost 40 T in size, so we are currently organizing our open source content. As the original images themselves are open source, we plan to organize the URLs of the original images and then open source the corresponding segmented images that we have processed along with structured information such as BLIP captions, masks, and boxes. Additionally, our code was publicly released four months ago and currently has 200+ stars.
>
> Q3-A3: Apologies for this. We had initially included the limitations in our first version but later removed them to control the content of the main body. We will make sure to add the limitations back and make the necessary adjustments.
> Regarding "failure case discussions," we will supplement them in Appendix G marked in red of the paper. Please refer to that section. Thank you for pointing it out.
> For the shortcomings of "editing attributes" and "rendering harmonious images with two subjects," we provide an example where the attributes related to the failed images are highlighted in red prompts. As for generating images with two subjects, if the provided image itself is missing one or both subjects, it may result in disharmonious final images.
>
> Q4-A4: Thank you for your reminder. The main text has been modified.
>
> Q5-A5: Thank you for your reminder. For inference, we utilize the PNDM scheduler for 50 denoising steps. We maintain a fixed text guidance scale of 3 and image guidance scale of 1.5 for all experiments conducted. These specifications will be highlighted in red in Appendix D marked in red of the paper.
>
>
> We are looking forward to your reply and happy to answer your further questions.
>
> Best regards
>
> Authors

---

### Official Review · Reviewer_83st · 2023-11-06

**Soundness:** 3 good
**Presentation:** 3 good
**Contribution:** 3 good
**Rating:** 6
**Confidence:** 4

**Summary:**

The authors have introduced "Subject-Diffusion," an innovative approach for personalized image generation that doesn't require test-time fine-tuning and only needs a single reference image. They created a large dataset and a unified framework that combines text and image information. The results show Subject-Diffusion outperforms other methods in generating single-subject, multi-subject, and customized images, marking a significant advancement in this field.

**Strengths:**

1.	They develop an automatic pipeline for constructing a substantial and well-organized training dataset, consisting of 76 million open-domain images and 222 million entities.
2.	Their work introduces a pioneering framework for personalized image generation, addressing the challenge of simultaneously generating open-domain personalized images for both single and multi-concept subjects, all without requiring test-time fine-tuning. This framework relies solely on a single reference image for each subject.
3.	The experimental results, both quantitative and qualitative, showcase the exceptional performance of their framework when compared to other state-of-the-art methods, confirming its effectiveness in personalized image generation.

**Weaknesses:**

1.	While the authors have access to a substantial training dataset containing subject information (segmentation, text descriptions, and bounding boxes), it appears that the method may not introduce sufficiently novel or distinctive techniques compared to previous approaches. Instead, it seems to be a combination or integration of existing methods.
2.	The training mechanism leverages multiple concept information from a single image, which is indeed a notable feature of the proposed model. However, it doesn't provide a detailed explanation or evidence to support the claim that this single-reference image approach can consistently yield better results compared to fine-tuning methods with access to multiple subject images.

**Questions:**

See the weakness section.

---

> ### Author Response · Authors · 2023-11-17
>
> W1: While the authors have access to a substantial training dataset containing subject information (segmentation, text descriptions, and bounding boxes), it appears that the method may not introduce sufficiently novel or distinctive techniques compared to previous approaches. Instead, it seems to be a combination or integration of existing methods.
>
> A1: Thank you for your review. We acknowledge that the paper focus more on engineering experiments and applications. Our primary motivation for this work is to alleviate the additional costs associated with fine-tuning that is typically required in current personalized generation methods. To achieve this goal, we have constructed a large-scale SSD dataset (which is currently being prepared for public release) and designed a unified framework that incorporates the strengths of many existing personalized generation methods. We would like to express our gratitude to the researchers who have worked on this topic before us. Moving forward, we will do our best to address these issues and aim to solve current personalized generation problems from a novel theoretical perspective.
>
> W2: The training mechanism leverages multiple concept information from a single image, which is indeed a notable feature of the proposed model. However, it doesn't provide a detailed explanation or evidence to support the claim that this single-reference image approach can consistently yield better results compared to fine-tuning methods with access to multiple subject images.
>
> A2: In fact, compared to fine-tuning methods with access to multiple subject images, our method does have advantages in fidelity-related metrics for multiple subject image generation. However, the diversity of our generated outputs may be weaker, which is also one of the major weaknesses of our paper. This is due to the fact that our approach is designed to prioritize the fidelity of the subject, which inevitably sacrifices the variability in generated outputs, such as the ability to edit attributes, etc. Balancing these two abilities remains a challenging problem in personalized generation and is an area where we aim to improve. Of course, this situation is in line with single-subject generation.
>
> We are looking forward to your reply and happy to answer your further questions.
>
> Best regards
>
> Authors

---

### Comment · Area_Chair_35UR · 2023-11-20

Dear reviewers,

As the Author-Reviewer discussion period is going to end soon, please take a moment to review the response from the authors and discuss any further questions or concerns you may have.

Even if you have no concerns, it would be helpful if you could acknowledge that you have read the response and provide feedback on it.

Thanks,
AC

---

### Meta-Review · Area_Chair_35UR · 2023-12-05

**Metareview:**

This paper proposes Subject-Diffusion for Personalized Text-to-Image Generation. The proposed model leverages additional control signals such as bounding box and segmentation mask to enhance performance. A large-scale SDD dataset is proposed. This paper receives mixed ratings of (3, 5, 6, 6), and no consensus has been reached during the discussion period. On the one hand, the reviewers appreciate the efforts of proposing of an approach that does not require test-time fine-tuning, and the proposed SDD dataset. On the other hand, the reviewers have concerns on novelty, diversity and quality of the outputs, and the lack of comparison to existing works. The AC have gone through the paper, reviews, and responses, and shares the same concerns as the reviewers. Therefore, a rejection is recommended.

**Justification For Why Not Higher Score:**

After reading the paper, reviews, and response, the AC shares the same concerns of the reviewers about the novelty, quality, and lack of comparison.

**Justification For Why Not Lower Score:**

N/A

---

### Decision · Program_Chairs · 2024-01-16

Reject